# Lookism, a Leak in the Career Pipeline? Career Perspective Consequences of Lookism Climate and Workplace Incivility

**DOI:** 10.3390/bs14100883

**Published:** 2024-10-01

**Authors:** Miren Chenevert, Cristian Balducci, Michela Vignoli

**Affiliations:** 1Department of Psychology and Cognitive Science, University of Trento, 38068 Rovereto, Italy; michela.vignoli@unitn.it; 2Department for Life Quality Studies, University of Bologna, 47921 Rimini, Italy

**Keywords:** lookism climate, workplace incivility, imposter syndrome, perceived employability, leaky pipeline

## Abstract

Despite strides toward gender equality in the workforce, women continue to face significant challenges, including the “glass ceiling” and the “leaky pipeline”, partially stemming from low occupational self-confidence. This study examined whether a climate of lookism leads to workplace mistreatment, undermining employees’ perceptions of job competence and career potential, with a focus on gender differences. Using a cross-sectional design, data from 699 Italian workers (42.8% male, 56.3% female) were analyzed through multi-group structural equation modeling. The model explored relationships between lookism climate, workplace incivility, imposter syndrome, and perceived employability. The results revealed a full serial mediation for women; lookism climate was positively related to workplace incivility, which in turn was related to imposter syndrome, negatively impacting perceived employability. For men, no serial mediation was found; lookism climate was directly related to both incivility and imposter syndrome, with no significant relationship between the two. Like women, men experienced a negative relationship between imposter syndrome and perceived employability, yet this relationship was stronger for men. This study highlights that identifying and addressing workplace climates that foster subtle mistreatment can prevent larger issues like the leaky pipeline, suggesting targeted organizational-level intervention and prevention strategies can enhance job competence perceptions and career potential for both genders.

## 1. Introduction

A simple search of #prettyprivilege provides you with over 12,000 posts on Instagram and over 38,000 videos on TikTok. These social media discourses on pretty privilege speak about the societal advantages of being physically attractive, particularly for women. Such advantages include anything from daily “perks”, such as getting free drinks at a bar, being offered help as soon as you walk into a store, or being allowed to skip lines at nightclubs, to more profound advantages, such as being selected for a job position, getting promoted, or even earning more. “Pretty privilege” is more than just a social media hashtag and personal anecdotes. In the scientific literature, the prejudice/discrimination based on a person’s physical appearance, particularly their attractiveness, has been termed “lookism” [1]. Research delving into the phenomenon of lookism and the “beauty premium” within work environments reveals a multitude of advantages enjoyed by attractive individuals, including enhanced evaluations for attractive job applicants, superior performance ratings, presumed leadership capabilities, increased opportunities for advancement and career progression, and even elevated wages [1,2,3]. While gender and lookism have been explored regarding selection and evaluation outcomes, the intersection of lookism, particularly as a workplace climate, and gender regarding internal outcomes remains understudied. Moreover, previous research in work and organizational psychology has primarily focused on the advantages enjoyed by attractive individuals and the disadvantages faced by those less attractive, viewing lookism as individual acts of discrimination. However, this perspective overlooks the broader impact lookism may have when it becomes embedded in a workplace climate that emphasizes physical appearance over competence indicators. Such a climate may negatively affect all employees, regardless of their attractiveness, by fostering an environment that values superficial traits over professional abilities. To address this research gap, Dossinger and colleagues [1] developed a scale to measure lookism as a workplace climate. The scale measuring lookism climate, defined as the collective perceptions within a workplace regarding the emphasis placed on employees’ physical attractiveness, enables researchers and organizations to understand how perceptions of physical attractiveness are embedded within the broader organizational context. The approach acknowledges that lookism is not merely an individual concern but a collective phenomenon that can impact group dynamics and organizational culture. As lookism climate has only recently been operationalized into a scale, empirical research exploring it as a workplace climate remains limited. Dossinger and colleagues’ [1] validation study investigated outcomes of lookism climate specifically, yet the authors continued to focus on individual workplace outcomes rather than the broader work environment. The validation study revealed that lookism climate is significantly correlated with increased work stress, work anxiety, and turnover intention, as well as decreased job satisfaction [1]. This study aims to look at the work environment as well as individual outcomes of lookism climate, exploring whether a climate driven by appearance-based discrimination has widespread implications by fostering a hostile work environment. This atmosphere may not only perpetuate appearance-based mistreatment but could also pave the way for other forms of mistreatment that impact all employees.

Much discourse surrounding the challenge of addressing appearance-based discrimination, which contributes to a lookism climate, highlights the difficulties in formulating laws and policies to combat lookism due to its subtle nature [3]. In contrast, more overt forms of discrimination have been more effectively addressed by organizations and policy makers alike. In the U.S., Title VII of the Civil Rights Act prohibits workplace discrimination based on various factors, including race, color, religion, sex, and national origin [4]. Similarly, in Europe, the Equal Treatment Directive [5] implements the principle of equal treatment between men and women in EU labor law. Additionally, to combat gender inequalities in the workplace, quota laws have been put in place, such as the “Quote rosa” in Italy, which refer to the number of positions reserved for women in the workforce of certain public and private institutions [6]. However, gender disparities in the labor market persist. While mandated gender quotas on corporate boards can increase female representation at the board level, the spillover effects on women in top executive or high-earning positions may be moderate and imprecisely estimated. This indicates that simply increasing female representation on boards may not directly translate into significant improvements in gender diversity in top leadership and high-earning roles within firms [7].

The European Commission notes several sources of remaining gender inequality in the workforce, highlighting sectoral segregation and the glass ceiling as the main contributors. Sectoral segregation and the glass ceiling are intertwined phenomena that spotlight the labor market restrictions for women that reflect gender bias stemming from centuries-old traditional gender roles. Sectoral segregation refers to the overrepresentation of women in relatively low-paying sectors, while the glass ceiling refers to the underrepresentation of women in top-level company positions across all sectors [8]. The concept of the leaky pipelines describes the process leading up to the glass ceiling, describing that there is a “leak” from the abundant female representation in lower-level positions to the scarce female representation in upper-level positions. As of 2019, only 6% of CEOs from companies listed on the Milan Stock Exchange were women, despite women making up 43% of corporate boards of directors [9].

The scholarly literature has delineated two principal explanations for the observed “leak” of female representation within the pipeline from lower-tier positions to higher-level roles, apart from the well-known “motherhood tax”. The first explanation centers on gender disparities in productivity. Studies have revealed that men tend to marginally outperform women on certain performance metrics, notably in areas such as overall productivity [10]. However, discerning whether these disparities stem from inherent gender traits or are shaped by societal and cultural pressures remains a challenge. The second explanation posits the reluctance of women to seek promotions. Previous research has indicated that women exhibit lower levels of self-assurance compared to men in professional settings, consequently diminishing their propensity to pursue career advancements [10]. However, what factors contribute to the prevailing pessimism among women regarding their career prospects, competencies, and capabilities?

Workplace injustice, mistreatment, and discrimination of any kind negatively impact employees and their perceptions of their occupational capabilities, despite discriminatory factors such as attractiveness and gender having no significant relationship with intelligence and competence [3,11]. In fact, Kaiser [12] found that covert workplace gender discrimination shapes women’s views on their career opportunities. Moreover, experiencing other types of subtle mistreatment, such as rude behaviors in the workplace (also known as incivility), is associated with lower self-esteem, self-image, self-efficacy, and self-confidence [13,14,15,16,17]. Thus, this study aims to explore the impact of a negative workplace climate characterized by appearance-based discrimination (lookism climate) on fostering workplace mistreatment that affects all employees, regardless of their physical appearance. The study will further examine how such mistreatment influences the self-perceptions of both women and men in their workplace and regarding their employability. The broader objective of the study is to investigate whether a lookism climate and workplace incivility account for leaks in the metaphorical organizational pipeline while also remaining open to the possibility that these processes may equally harm men. The results of this study will reveal whether organizational factors, such as lookism climate, contribute to further mistreatment, thereby identifying key areas for targeted interventions. Additionally, the study will assess gender differences in the effects of lookism climate and workplace mistreatment, providing crucial insights for developing tailored employee resources and support systems.

## 2. Theoretical Background

In the realm of occupational psychology, lookism emerged initially as a form of mistreatment and more recently as an underlying workplace climate. In organizational settings, lookism refers to the preferential treatment of physically attractive employees in comparison to those deemed less attractive. Consequently, lookism can be perceived as appearance-based discrimination or “aesthetic injustice” [2,18]. Noted as a form of neo-discrimination, lookism can be challenging to prove and thus difficult to address within legal frameworks despite having similar consequences to more overt discrimination [19]. Discourse on lookism as appearance-based discrimination typically refers to “taste-based” lookism, where decision-makers prefer attractive individuals based solely on personal biases or preferences, without any merit-based justification. However, evidence suggests the existence of “statistical lookism”, in which decision-makers assume that certain traits, like attractiveness, are correlated with other desirable qualities such as competence or productivity. This perception may be partly rooted in fact. For example, “pretty privilege” in society reflects that attractive individuals often benefit from increased social capital—a valuable asset in the workplace. Attractive workers may, in fact, possess stronger interpersonal skills, which can lead to better networking, enhanced productivity, and improved overall performance. Additionally, evidence suggests a slight advantage for attractive individuals in human capital, although this advantage often reflects the benefits of taste-based lookism itself. Attractive individuals may have better access to educational and professional opportunities, which can enhance their skills and qualifications [20]. The notion of lookism climate pertains to individual or collective perceptions that a workplace implicitly or explicitly prizes employee physical attractiveness [1]. A lookism climate encompasses both individual and collective perceptions regarding the value placed on physical appearance within an organization, illustrating how shared beliefs about physical attractiveness can shape the overall workplace environment. Like other climate constructs, lookism climate emerges from individuals observing and interpreting social cues and events over time, such as lookism manifesting in hiring practices, promotions, and daily interactions among colleagues. These interactions reveal how attractiveness is (or is not) evaluated and rewarded within the workplace—an aspect not included in other climate constructs. This focus on social cues and observed behaviors suggests that lookism climate is primarily aligned with shared perceptions of taste-based lookism—where attractiveness is preferred regardless of any connection to professional qualities. The scale items measuring lookism climate do not indicate that organizations assume attractiveness to correlate with competence, productivity, or other job-related attributes, as would be the case with statistical lookism.

In their scale development and validation study, Dossinger and colleagues [1] found that lookism climate significantly affects employees’ focus on grooming and appearance, a trend not found in other climate contexts. Employees may invest more time and resources into their appearance in response to perceived expectations, leading to this distinctive behavioral outcome, which can be viewed as an additional job demand. Additionally, Dossinger and colleagues [1] identified another key characteristic of lookism climate: the tendency for employees to engage in social comparisons based on appearance, which can result in feelings of inadequacy, resentment, jealousy, or competition in and out of the workplace. Indeed, lookism climate interacts with other workplace climates, such as competitive climates, which similarly reward individual attributes based on comparisons with others [1,21]. However, lookism climate is distinct in that it focuses on appearance—a more fixed and relatively unalterable criterion compared to attributes like performance and productivity that are emphasized in competitive climates [1]. While individuals can take steps to align their appearance with organizational standards—often influenced by societal norms, such as losing weight, altering their hairstyle, or dressing according to current fashion trends—some physical characteristics remain immutable. This can lead to increased frustration for those who are perceived as less attractive. Conversely, individuals who are considered attractive may feel objectified, perceiving that their value to the organization is solely based on their physical appearance. This objectification not only dehumanizes them but can also heighten anxiety about maintaining their appearance [1,22].

Organizational climates play a pivotal role in shaping employee attitudes, behaviors, and outcomes, creating a shared understanding among employees regarding what is rewarded, supported, expected, and tolerated within an organization. This shared understanding significantly influences workplace dynamics and perceptions involved in the occurrence of counterproductive workplace behavior and mistreatment [21]. Indeed, a meta-analysis of the antecedents and consequences of workplace incivility over 20 years identified organizational climate as a primary antecedent of the occurrence of workplace incivility. The study found that organizational climates can either promote respectful interactions or inadvertently condone uncivil behavior [23].

Organizational climates associated with lookism climate—such as competitive climates (positively correlated), climates of perceived justice, and climates of inclusion (negatively correlated)—significantly influence employee career trajectories and work-related processes and outcomes, including adverse effects on interpersonal relationships [1,24]. Indeed, the theory of organizational justice describes that employees have a fundamental expectation of considerate treatment in the workplace, and employees’ perceived violations of this expectation is key to understanding individual actions and reactions in the workplace [25]. Lookism climate, which places value on employee physical attractiveness regardless of correlated capabilities, demonstrates poor distributive, interactional, and even procedural justice within an organization. A climate of lookism undermines the principles of organizational justice by creating a systemic environment where individuals are judged and treated based primarily on their appearance rather than their skills, qualifications, or contributions. Indeed, the existing literature suggests that many people consider an emphasis on physical appearance in the workplace as unjust and morally questionable [26]. This emphasis on appearance and consequential differential treatment undermines the principle of distributive justice, which advocates for the fair allocation of resources and opportunities based on merit rather than appearance alone. Furthermore, a lookism climate can perpetuate stereotypes related to appearance, reinforcing inequalities within the organization and promoting statistical lookism, or “justifiable” appearance-based discrimination. This perpetuation of stereotypes often results in a lack of respect and recognition for individuals who do not fit the idealized image, thereby violating the principle of interactional justice, which emphasizes respectful and dignified treatment in interpersonal interactions. Additionally, a lookism climate fosters an exclusive environment where only those meeting specific attractiveness criteria feel valued and included, contradicting the principles of procedural justice that advocate for fair and inclusive decision-making processes. Employees who feel marginalized due to their appearance may become disengaged, affecting overall workplace morale. As previously discussed, violations of organizational justice within a lookism climate can also be perceived by the employees that lookism climate is deemed to “favor”. Lookism climate, based on perceptions of taste-based lookism, undermines the meritocratic principles that organizations often strive to uphold. When physical appearance becomes a significant factor in evaluations and decision-making, it shifts the focus away from skills, qualifications, and performance. This shift can create a culture where attractive employees feel that their hard work and contributions are overshadowed by their superficial attributes. Consequently, attractive employees within the “ingroup” may struggle to achieve self-actualization in the workplace due to their appearance superseding the importance of their personal growth, fulfillment, and potential [27]. Additionally, these employees might experience collective guilt, which involves the reluctant acceptance of the ingroup’s “misdeeds”, in the case of lookism climate, taste-based preferential treatment. These cumulative lookism climate consequences can lead to cognitive dissonance and anxiety for employees traditionally viewed as being “privileged” by a lookism climate [28]. Research shows that perceived injustice, such as unfair treatment, favoritism, or lack of transparency, can lead to frustration, resentment, decreased job satisfaction, and disengagement, which may result in perpetrating and experiencing uncivil behaviors [29,30].

Moreover, lookism climate fosters social comparisons among employees, resulting in individualistic behaviors, diminished collaboration, and even deviant workplace behaviors [1]. In an unjust workplace climate, such as that of lookism climate, employees compare their own justice experiences with those of their coworkers, leading to feelings of envy and depletion of self-regulatory resources. The self-regulatory perspective posits that individuals strive for a desired internal state by assessing the gap between their actual state and reference points. In the context of social justice comparisons, discrepancies in perceived fairness among coworkers can trigger emotional responses and affect employees’ self-control resources and behaviors, thus leading less attractive employees to enact uncivil behaviors [31]. Alternatively, less attractive individuals feeling marginalized and excluded in their work settings might cope internally and withdraw socially or experience declines in productivity and performance, rather than lashing out externally. This internalization process can exacerbate the mistreatment they encounter, as it may be perceived as justified or unlikely to elicit retribution or formal reporting [32,33]. Additionally, attractive employees who benefit from the social capital of a lookism climate and feel guiltless may feel emboldened to engage in interpersonal mistreatment against the outgroup, viewing it as condoned by the organization or as a part of their elevated status in the workplace [33,34]. Therefore, we expect that lookism climates will foster incivility experiences for all employees in the workplace, and we posit the following hypothesis:

**Hypothesis** **1.***Lookism climate will positively relate to workplace incivility experiences*.

In the past, most research on workplace mistreatment focused on intense and explicit forms of workplace mistreatment, such as workplace bullying and sexual harassment. Thanks to this research, laws such as convention 190 in Italy, against violence and harassment in the workplace, have been implemented to address these serious forms of workplace mistreatment [35]. Yet today, the focus has turned to more subtle types of workplace mistreatment. In fact, in light of a 1997 article [36], which suggested that the majority of workplace mistreatments are more subtle, Andersson and Pearson [37] proposed a new concept: workplace incivility. Unlike workplace bullying, which is composed of high-intensity and high-intent negative acts by the same individual(s) towards the same individual(s) repeatedly over time, workplace incivility encompasses lesser forms of mistreatment in the workplace with an ambiguous intent to cause harm and no specific timeframe. These behaviors could include something as simple as leaving the copier jammed for the next person, sending a passive aggressive email or raising one’s voice at a colleague. In this way, workplace incivility can be considered to be rude or discourteous behaviors that show a lack of respect for others in the work environment [38].

Despite the covert and seemingly benign nature of workplace incivility, studies have found that experiencing workplace incivility has similar negative outcomes to experiencing workplace bullying and other intense and intentional forms of workplace mistreatment [39,40,41,42]. Bandura’s theory of self-efficacy and Cooley’s looking-glass self-theory both suggest that our self-efficacy beliefs and self-perceptions are influenced by how others view us. Indeed, when one is belittled and socially undermined by others, key elements of workplace incivility, one’s beliefs about their ability to succeed in a specific context significantly decrease [43,44], These negative beliefs about one’s occupational abilities and competence can make one feel as though they do not deserve to be in their workplace, a phenomenon known as imposter syndrome. Impostor syndrome, also known as impostor phenomenon, fraud syndrome, perceived fraudulence, or impostor experience, is a psychological pattern, typically investigated in academic or workplace settings, in which individuals doubt their accomplishments and have a persistent fear of being exposed as a “fraud” [45]. A systematic review of imposter syndrome found that fears of fitting in and maintaining a social standing within the work environment is a main predictor of experiencing imposter syndrome [45]. Experiencing undermining, hostile, or uncivil behaviors can result in social estrangement and lead one to feel out of place, disliked, and less competent. Indeed, the just world theory and the attribution theory explain that individuals who face mistreatment, such as incivility, often internalize the mistreatment event and see themselves as deserving of the incivility due to their lack of skill, competence, or likeability [43]. Furthermore, a review focused on workplace contexts and women’s career trajectories found that positive treatment from colleagues can help alleviate impostor feelings, while negative treatment can exacerbate these feelings [24]. Thus, experiencing incivility may foster this “imposter” sentiment by breaking down our innate need for competence and relatedness and deepen sentiments of self-doubt. Accordingly, we developed the following hypothesis:

**Hypothesis** **2.***Workplace incivility will positively relate to imposter syndrome*.

Perceived employability refers to an individual’s belief in their ability to obtain and maintain employment within the job market. This concept encompasses five dimensions: knowledge and skills, capacity for learning, mastery of career management and job search, professional knowledge, and resilience and personal efficacy [46]. Imposter syndrome chips away at the positive self-beliefs that compose the dimensions of perceived employability. As mentioned previously, imposter syndrome can undermine self-efficacy beliefs, a crucial dimension of perceived employability. In addition, resilience, or the ability to bounce back from setbacks and challenges, is even more challenging for those who already feel like they are a fraud in their work environment. Furthermore, a systematic review found that impostor syndrome is associated with impaired job performance, job satisfaction, and burnout among various employee populations, which may impact the self-perceived employability component of knowledge and skills and capacity for learning [45]. Indeed, researchers have referenced the conservation of resources theory, proposing imposter syndrome as a self-inflicted demand, which leads to emotional exhaustion and feelings of hopelessness when it comes to reaching an adequate level of knowledge and capacity [47]. Imposter syndrome can also affect the perceived employability component of master of career management and job search. Experiencing imposter syndrome can lower one’s propensity to engage in professional development activities such as continuing education, skill-building, and seeking opportunities for growth and learning. Individuals grappling with imposter syndrome may feel unworthy of investing time and effort in their professional development due to persistent feelings of inadequacy, self-doubt, and fear of being exposed as a fraud. The negative impact of imposter syndrome on professional development can create a barrier to enhancing competence and self-assurance in one’s professional role. This reluctance to engage in growth opportunities may stem from a belief that no amount of learning or skill-building can alleviate the perceived fraudulence or lack of expertise associated with imposter syndrome [48]. Furthermore, individuals with imposter syndrome may employ maladaptive coping mechanisms to manage their anxiety and self-doubt. These coping strategies, such as over-preparing, procrastinating, self-sabotage (self-fulfilling prophecy), and maintaining a low profile may lead to missed opportunities, avoidance of challenges, and increased stress and burnout [49]. Therefore, failing to engage in professional development can diminish one’s confidence in their career management and job search abilities. In conclusion, imposter syndrome significantly undermines multiple dimensions of perceived employability, diminishing self-efficacy, resilience, engagement in professional development, and career management, ultimately lowering an individual’s holistic perception of their employability. Consequently, we hypothesize the following:

**Hypothesis** **3.***Imposter syndrome will negatively relate to perceived employability*.

In 1990, Acker introduced her theory of gendered organizations, asserting that gender is a fundamental organizing principle within organizational structures and processes [50]. Since then, numerous studies have employed Acker’s theory to investigate how gender influences inequality in organizations and the structural segregation of women. In a recent review, Bates [51] critiqued the past applications of Acker’s theory for being prone to confirmation bias. Bates suggested adopting an abductive approach in future research to remain open to unexpected findings that challenge existing assumptions about gender in organizations. Given that lookism, workplace mistreatment, and imposter syndrome impact men and women differently, we expect to uncover gender differences within our model. Following Bates’ [51] recommendations for future research on gender and organizations, we employed an abductive approach to explore how these gender differences manifest within the hypothesized model.

Lookism presents different constraints for men and women. Research suggests that attractive individuals, regardless of gender, are more likely to secure employment, promotions, and higher pay throughout their careers, with facial attractiveness significantly influencing occupational status for both men and women [52]. However, one study in the Italian population found that while multiple aspects of facial attractiveness are required to predict favorable hiring decisions for female applicants, only facial competence is needed for male applicants, indicating a distinct double standard for women [53]. This double standard extends beyond facial attractiveness. For instance, women wearing form-fitting and “sexy” clothing are judged as less competent than those wearing loose, ill-fitting clothing [54,55,56]. These double standards exemplify the “erotic capital” that women wield in society, which entails a precarious balancing act in the workplace known as “aesthetic labor”. For women, aesthetic labor requires finding a balance between femininity and professionalism in order to be taken seriously [57]. While men may not face the fatigue of high aesthetic labor, they also do not benefit from the “perks” of possessing erotic capital. Male physical attributes determining attractiveness are more rigid than those for females, with women having more control over their appearance through makeup, hairstyling, and fashion choices—an advantage not shared by men [3,52]. Furthermore, while imposter syndrome can affect anyone, research has found that it disproportionately affects women, underrepresented minorities, individuals from low socioeconomic backgrounds, and early career workers [48,49]. Like the just world theory, the objectification theory highlights how women are taught to internalize an observer’s perspective as their view of themselves and their worth. Stigma can reinforce feelings of inadequacy and self-doubt, amplifying the imposter phenomenon and creating a cycle of negative self-perception. Indeed, stigmatized populations, such as women, often experience higher levels of imposter syndrome, which can be exacerbated by workplace mistreatment [58]. These gendered social learnings, paired with the traditionally masculine workplace environment, suggest that the self-esteem effects of experiencing workplace incivility may differ by gender. Lastly, while existing research indicates that women experience higher levels of imposter syndrome with respect to men, the differential effects across genders are not well understood. Although women may report stronger feelings of imposter syndrome, this does not necessarily correlate with greater negative impacts. The leaky pipeline phenomenon and women’s reluctance to seek promotions suggest that imposter syndrome may significantly undermine women’s perceived employability. Conversely, traditional gender roles may exert additional pressure on men to excel in the workplace, potentially amplifying the effects of imposter syndrome. Thus, this study aims to explore how gender dynamics influence the occupational self-perceptions of men and women with the following exploratory hypothesis:

**Hypothesis** **4.***Gender differences will be present in the serial mediation model*.

Positive supervisory behaviors such as support and integrity have been found to positively and directly influence employee well-being and job attitudes. In addition, the way in which managers organize and manage their employees influences the cooperation, stress levels, and effective coping styles of their subordinates when facing stressful events such as working in a climate of perceived appearance-based discrimination [59]. Managerial competencies that have been found to have a positive link to the stress management and well-being of employees can be placed into three categories: altering working conditions, altering environment transaction, and managing individuals within a team [60]. The first competency describes the way in which managers can help employees cope with workplace stress by removing obstacles such as work overload, isolation, lack of autonomy, and isolation in order to create the optimal work environment for the employee. The second competency involves the way in which a manager helps an employee cope with stress by improving their interaction with their work environment. This includes actions such as providing employees with services like employee assistance programs, links to stress management resources, training for behavioral skills, and meditation and relaxation techniques, all to relieve the physical and psychological effects of stress. The last competency, particularly relevant to avoiding interpersonal mistreatment, involves how a manager adequately handles inter-team conflict in the work setting. This includes skills such as strategic development in tension reduction and proper allocation of individual tasks within a team [60]. Additionally, competent managers exhibit fair treatment and clear communication and provide employees with training and development opportunities and proper resource allocation. By focusing on these aspects of leadership and management, managers can positively influence employee perceptions of organizational justice, which, in turn, can enhance employee commitment and engagement within the organization and improve colleague relationships [61]. Competent managers who augment the poor work environment, such as one where lookism climate exists, offer their subordinates social support, promote team cooperation, and adopt zero-tolerance attitudes towards mistreatment, and they may be able to buffer the consequences of a negative organizational climate. Indeed, the social learning and social exchange theories suggest that if managers model appropriate and ethical behaviors, those closest to them will adopt these behaviors as well [62]. Past research has found that competent and ethical managers moderate the positive relationship between both workplace stressors and individual predispositions and workplace mistreatment experiences, including workplace incivility and bullying [63,64]. Thus, we expect the following hypothesis:

**Hypothesis** **5.***Managerial competencies will weaken the relationship between lookism climate and workplace incivility*.

## 3. Materials and Methods

### 3.1. Study Design and Procedure

To test the hypotheses, this study utilized a cross-sectional design to test each path of the multi-group serial mediation between the independent variable, lookism climate; first mediator, workplace incivility; second mediator, imposter syndrome; and dependent variable, perceived employability. In addition, the moderating effects of managerial competencies on the path between lookism climate and workplace incivility were tested. The distributed questionnaire established a general socio-demographic profile of the participants, including variables such as Italian region of work, BMI, working hours, and working modality in addition to measures for the control variables age and ratings of self-perceived attractiveness. Lastly, the questionnaire contained measures for accessing the variables of interest: lookism climate, workplace incivility, imposter syndrome, perceived employability, and managerial competencies. Data collection took place through an online survey between February and March. Participant criteria included adults over the age of 18, currently employed for six months or more, working at least 25 h per week, and with a good knowledge of the Italian language. Informed consent was obtained from all subjects involved in the study.

### 3.2. Participants

Data collection through snowball sampling ended up with 1023 survey responses. Following data cleaning by applying the methods advised by Ward and Meade [65], the final sample included 705 Italian workers. Out of the 1023 survey responses, only 773 were complete responses. Of the complete responses, 40 were flagged as duplicate responses and 8 as potential bots by Qualtrics’ quality check and were excluded. Lastly, 20 participant responses failed at least one of the three attention checks included in the questionnaire and were thus excluded. Of the final sample of 705 participants, 56.3% were women (N = 397), 42.8% were men (N = 302), and 0.01% (N = 6) were gender queer. Due to the low sample size of gender queer workers, and for the purpose of the multi-group analysis, the gender queer participants were excluded. The average age of the participants was 38 years old, with the youngest participant being 18 and the oldest 67; 77.4% (N = 546) of the participants worked in northern Italy and 88% (N = 623) of participants rated themselves as averagely and above averagely attractive. Participants had an average of 11 years of experience in their respective sector and worked on average 38 h per week, with most participants (79.1%, N = 558; 17.7%, N = 125) working in-person or hybrid, respectively. Table A1 in Appendix A depicts the personal and work-related characteristics of the sample by gender.

### 3.3. Measures

#### 3.3.1. Lookism Climate

Lookism climate was measured using the six-item Lookism Climate Scale developed by Dossinger and colleagues [1]. The scale is unidimensional with an original reliability coefficient of 0.94 [1]. In this study, the internal reliability of the scale was 0.87. The introduction to the scale reads “Think about your workplace in the last six months and indicate your level of agreement with the following statements”. Participants read each of the following statements (e.g., “Being physically attractive is highly valued in my workplace”) and indicated their level of agreement on a five-point Likert scale (1 completely disagree–5 completely agree).

#### 3.3.2. Workplace Incivility

Workplace incivility was measured using the seven-item Workplace Incivility Scale developed by Cortina et al. [66]. The scale is a validated unidimensional scale with a reliability coefficient of 0.89 [66]. In this study, the internal reliability of the scale was 0.87. The introduction to the scale reads “In the last six months, how often has a colleague/superior enacted one of the following behaviors in the workplace”. Participants read each behavioral description (e.g., “Addressed you in unprofessional terms, either publicly or privately?”) and expressed the frequency of experience on a five-point Likert scale (1 never–5 always).

#### 3.3.3. Imposter Syndrome

Impostor syndrome was measured using 7 items from the original 20-item Impostor Phenomenon Scale developed by Clance [67] and later validated by Chrisman and colleagues [68]. The reliability coefficient of the validated scale was 0.92; for this study, the reliability coefficient was 0.80. The introduction to the scale reads “Think about your work situation in the last six months, for each of the following statements, indicate your level of agreement”. Participants read each of the following statements (e.g., “I can give the impression that I’m more competent than I really am.”) and indicated their level of agreement on a five-point Likert scale (1 completely disagree–5 completely agree).

#### 3.3.4. Perceived Employability

Perceived employability was measured using the eleven-item Self-perceived Employability Scale [46]. The scale was validated unidimensionally and bi-dimensionally. The reliability coefficient of the unidimensional scale is 0.83 [46]. In this study, the internal reliability of the unidimensional scale was 0.78. The introduction to the scale reads “Read the following statements and indicate your level of agreement with each statement”. Participants read each of the following statements (e.g., “The skills I have gained in my present job are transferable to other occupations outside this organization.”) and indicated their level of agreement on a five-point Likert scale (1 completely disagree–5 completely agree).

#### 3.3.5. Managerial Competencies

Managerial competencies were measured using the nine-item Stress Management Competency Indicator Tool developed by Toderi and Sarchielli [60]. The original scale had a unidimensional reliability measure of 0.85 [60]. In this study, the internal reliability of the scale was 0.90. The introduction to the scale reads “Read the following statements and indicate your level of agreement with each statement”. Participants read each of the following statements (e.g., “My supervisor makes it explicit that he will take ultimate responsibility if things go wrong.”) and indicated their level of agreement on a five-point Likert scale (1 completely disagree–5 completely agree).

#### 3.3.6. Control Variables

The confounding variables of age and ratings of self-perceived attractiveness that may influence perceptions of lookism climate, experiences of mistreatment, and constructs relating to self-esteem and self-efficacy (e.g., imposter syndrome and perceived employability) were controlled for. Self-perceived attractiveness was controlled for in order to assess the overall implications of lookism climate. Participants reported their age as well as a rating of their self-perceived attractiveness on a scale from 1–10, spanning from least attractive (1) to most attractive (10). Although self-perceived attractiveness may be subject to bias, previous research on lookism and lookism climate has demonstrated that an individual’s perception of their own attractiveness is a more significant predictor of outcomes related to lookism and lookism climate than external assessments of attractiveness [3].

### 3.4. Data Analysis

Assumption tests for the moderated serial mediation model, the reliability analysis, the correlations, and the descriptive statistics of the sample were tested using the software IBM SPSS Statistics 22.0. A composite score for each variable (i.e., lookism climate, workplace incivility, imposter syndrome, perceived employability, and managerial competencies) was computed by adding the respective items of each scale and computing an average score, as each scale has been shown to be unidimensionally valid in previous studies. To evaluate the hypotheses, Mplus was used to conduct a multi-group, serial mediation model using structural equation modeling and the MLR estimator to account for assumption violations. Firstly, a multi-group SEM approach was adopted because it enables variances and covariances among factors and error variances to vary across groups, which was logically expected in this context. By accounting for these differences in a multi-group analysis, we can minimize the risk of biased estimates resulting from model misspecification [69]. Additionally, a multi-group SEM was utilized based on the recommendations of Cheung et al. [70], who suggest that a multi-group analysis is more appropriate when the moderating variable is categorical and moderates the relationship between continuous latent variables. The multi-group analysis was conducted to test the relationship between lookism climate and perceived employability through exposure to workplace incivility and consequently imposter syndrome. To evaluate possible moderation effects on the resulting serial mediation model for women, PROCESS was used, considering managerial competencies as a possible moderator. The accepted established significance level for this study was *p* < 0.05 for all the analyses conducted.

## 4. Results

First, means, standard deviations, and correlations between variables were calculated, as shown in Table 1.

### 4.1. Model Fit

Due to strong covariances between same-scale items, the parcel method was implemented, creating parcels based on the results of the model modification indices. After the implementation of parcels, despite a significant chi-square result, the other measurement model indices were within the recommended criteria established by Bentler, which includes CFI values above 0.90, TLI values above 0.90, RMSEA values less than 0.08, and SRMR values less than 0.08 (χ^2^ = 345.05 [female contribution = 202.20; male contribution = 142.85], df = 144, *p* < 0.001, CFI = 0.93, TLI = 0.91, RMSEA = 0.06, SRMR = 0.07) [71]. Table A3 in Appendix A shows the standardized factor loadings of each parcel for men and women as well as the standard error and two-tailed *p*-value. All parcels loaded strongly onto their respective latent construct. The first parcel, measuring employability, had the weakest factor loading among all the parcels; however, the value (0.38) still meets the cut-off of 0.30 proposed by MacCallum et al. [72] to demonstrate practical significance in a sample of 350 or more. Additionally, the factor loading of said parcel was equivalent for men and women, demonstrating measurement invariance between genders.

### 4.2. Serial Mediation: Multi-Group SEM by Gender Binary

The results of the multi-group structural equation modeling (SEM) analysis comparing men and women revealed gender differences in the serial mediation model. For women, lookism climate significantly predicted workplace incivility (β = 0.30, *p* < 0.05), which, in turn, significantly predicted imposter syndrome (β = 0.21, *p* < 0.05). Imposter syndrome, in turn, significantly predicted lower perceived employability (β = −0.22, *p* < 0.05). The direct effect of lookism climate on perceived employability was not significant, nor were any of the other paths, indicating a complete serial mediation for women with a significant negative indirect effect (−0.01, 95% CI [−0.02, −0.002]). For men, the path coefficients revealed that lookism climate significantly predicted workplace incivility (β = 0.25, *p* < 0.05); however, experiencing workplace incivility did not predict imposter syndrome (β = 0.10, *p* = 0.17). However, imposter syndrome did significantly predict lower perceived employability (β = −0.57, *p* < 0.05). Additionally, lookism climate directly predicted imposter syndrome (β = 0.15, *p* < 0.05). The direct effect of lookism climate on perceived employability was not significant. Thus, for men, there was no serial mediation, rather a single linear regression between lookism and workplace incivility and a full, non-serial mediation between lookism climate, imposter syndrome, and perceived employability with a significant negative indirect effect (−0.04, 95% CI [−0.09, −0.01]). All path coefficients for both men and women can be seen in Figure 1 below.

Measurement invariance for MLR model comparison was conducted in order to proceed with a comparison of the serial mediation models between genders. There were no significant differences in the configural, metric, scalar, and partial scalar measurements, indicating that the latent constructs were measured equivalently across groups. We then proceeded to test for structural path differences in the serial mediation model between women and men using the Satorra–Bentler scaled chi-square difference test; the results are depicted in Table A4 in Appendix A. The test revealed significant differences between men and women on the path between imposter syndrome and perceived employability (*p* < 0.05), with men having a notably stronger negative relationship between imposter syndrome and perceived employability compared to women.

### 4.3. Moderation Analysis

After a serial mediation model was found only for women, managerial competencies were tested as a moderator to investigate whether the strength of the relationship between lookism climate and workplace incivility depends on the competence of one’s manager. The analysis found a statistically significant moderated serial mediation, with the moderator managerial competencies reporting an index of 0.01 and bootstrapping confidence intervals between 0.0001 and 0.032 and a coefficient of −0.25 (*p* < 0.05). The R^2^; change showed that 2% of the variance in the indirect effect can be explained by the interaction between lookism climate and managerial competencies. The bootstrapping confidence intervals revealed that the moderation effect was only significant for the values of managerial competencies one standard deviation below the mean (95% CI [0.22, 0.42]) and at the mean (95% CI [0.13, 0.28]) but not at one standard deviation above the mean (95% CI [−0.02, 0.19]). In sum, the positive relationship between lookism climate and workplace incivility was weakened for those with adequately competent managers compared to those with incompetent managers. A visual representation of the moderating effects of managerial competencies in the serial mediation model for women can be seen in Figure 2 below.

## 5. Discussion

### 5.1. Study Contribution

The present study aimed to investigate the effects of lookism climate on perceived employability through the mediating roles of workplace incivility and imposter syndrome, with a focus on gender differences. The findings provide valuable insights into how this process differs for men and women, underscoring the intricate dynamics within workplaces and the critical role of managerial competence in shaping employees’ perceptions of their occupational abilities even under a negative work climate. This study contributes to the literature on the leaky pipeline and glass ceiling by demonstrating that negative workplace climates (such as lookism climate) and mistreatments (such as incivility) may contribute to women’s negative perceptions of their occupational abilities and employability perceptions, which may, in turn, influence their career advancement decisions. Furthermore, this study highlights the detrimental association between imposter syndrome and men’s perceived employability, emphasizing the importance of providing support to all employees, irrespective of gender, as men may not readily seek help. The absence of a significant direct effect in both the female and male models underscores the ripple effect of lookism climate on employees’ perceived employability. The results of this study suggest that lookism climate influences perceived employability indirectly through the negative workplace dynamics and sentiments it fosters. However, these findings should be interpreted with caution, as the cross-sectional design of the study prevents any causal conclusions from being made. Lastly, this study reinforces existing findings that effective leadership and competent management are essential for promoting employee well-being and fostering positive workplace environments for the prevention of workplace mistreatment. The results of this study suggest several organizational targets for mitigating mistreatment effects as well as preventing subtle workplace mistreatment occurrences, which may spiral into larger issues: inclusive workplace climates, well-trained managers, and gender-inclusive resources to boost employee confidence.

### 5.2. Gender Differences in Mediation Model: Consequences of Patriarchal Pressures

For women, the serial mediation model demonstrated that lookism climate significantly influences workplace incivility, which subsequently influences imposter syndrome, ultimately influencing lower perceived employability. The absence of a direct effect of lookism climate on perceived employability, combined with the significant indirect effect, underscores the importance of considering intermediate psychological and behavioral processes when examining the impact of lookism climate. This complete serial mediation suggests that lookism climate indirectly diminishes women’s perceived employability through a cascading effect of mistreatment and heightened imposter syndrome. This cascade from negative experiences to psychological impacts reveals one reason why women might be reluctant to apply for career advancements, contributing to the leaky pipeline and glass ceiling phenomena. The stronger path for men between imposter syndrome and perceived employability suggests that other mistreatment consequences, beyond imposter syndrome, may more significantly affect women’s perceived employability. Understanding these intermediate processes is crucial to addressing the barriers women face in advancing their careers, thereby tackling the root causes of the leaky pipeline.

In contrast, for men, the analysis did not support a serial mediation model. While lookism climate significantly influenced workplace incivility, this incivility did not significantly influence imposter syndrome. Instead, lookism climate directly influenced imposter syndrome, which in turn affected perceived employability. These findings suggest that the lookism climate scale may measure both lookism climate and lookism as a direct form of discrimination. No significant differences between men and women were found for indicators of attractiveness measured by the study, including Body Mass Index (BMI), self-rating of attractiveness, and body image. These results imply that the direct effect of lookism climate on imposter syndrome for men was not a result of a less or more attractive sample of men compared to women. Instead, the results of this study suggest that while lookism climate negatively affects women’s occupational self-perceptions by promoting experiences of further workplace mistreatment, lookism can immediately trigger negative self-perceptions in men.

The primary gender difference identified in this study, and the only statistically significant difference in the model between men and women, was that men exhibited a stronger relationship between imposter syndrome and lower perceived employability compared to women. The results offer insights into the negative cross-gender effects of patriarchal principles in the workplace, as these gender differences may be attributed to socialized gender roles and expectations in the workplace. The negative impact of traditional gender role expectations on men, particularly in the workplace, has been an undervalued area of research and general discourse. In 2023, Stanaland and colleagues [44] introduced the expectancy–discrepancy–threat model of masculine identity, which explains how perceived discrepancies between rigid masculine ideals and the actual self can trigger anxiety, shame, and even aggression in men. Men with imposter syndrome may feel their perceived fraudulence violates expected norms of dominance and power in the workplace, while women with imposter syndrome may expect these feelings, aligning with the prescribed female role of passive outsiders in the work environment. Indeed, this “underdog mentality”, “golem effect”, or, more colloquially, the “nothing to lose effect” may buffer the negative effects of imposter syndrome for women [73]. Opposingly, men with imposter syndrome may experience more severe outcomes of imposter syndrome, such as low perceived employability, as a consequence of the gender role violation imposter syndrome represents. These findings underscore the need to dismantle the traditionally masculine work environment and ensure that performance evaluations and opportunities are not influenced by gendered traits.

The literature has suggested that it is organizational underpinnings, not individuals, that foster the persistence of the masculine default at work [74]. Thus, organizations should seek to implement protocols and procedures that aim to steer away from expecting and rewarding masculine traits over female ones. Promoting a more inclusive workplace climate, as highlighted by this study, can improve employees’ self-perceptions of their occupational abilities, even in the face of mistreatment, regardless of gender. Decades of workplace inclusion research endorses protocols and procedures such as employee resource groups (ERGs), conflict resolution procedures, accountability systems (in line with the CPR model), bias training, diverse leadership models, inclusive performance metrics, and feedback and development opportunities [75]. The findings of this study highlight the importance of creating a gender-inclusive work climate for the well-being of both male and female employees, even when confronted with subtle workplace mistreatments.

### 5.3. The Importance of Managerial Competencies for Female Employees

Further analysis for women revealed that managerial competencies moderated the relationship between lookism climate and workplace incivility, as hypothesized. Specifically, the impact of lookism climate on workplace incivility was diminished when managers were adequately competent compared to when managers were incompetent. This finding underscores the critical role of competent management for female employee well-being, particularly by mitigating the adverse effects of lookism climate on workplace dynamics. The significant moderation effect of managerial competencies suggests that effective leadership can buffer against outcomes associated with negative workplace climates and workplace mistreatment that, in turn, influence women’s occupational self-perceptions. This finding is in line with the model of antecedents of the glass ceiling developed by Elacqua and colleagues [76], which highlights three interpersonal factors: mentoring, the existence of an informal network of senior managers, and friendly relationships with company decision-makers, as key to career advancement for women. Indeed, a recent study employing this model found that women who felt supported by their managers perceived more opportunities for promotion and a greater likelihood of being promoted, signaling that these women had positive perceptions of their employability [77]. The findings of this study not only support Elacqua and colleague’s [76] model but also suggest that managerial competencies may play a central role, rather than a merely moderating one, in diminishing workplace mistreatment experiences. Thus, enhancing managerial competencies through training and development programs could serve as a protective factor, reducing the incidences of workplace incivility and its subsequent negative effects on employee perceptions and behaviors. Indeed, a meta-analysis on leadership and workplace mistreatment found that change-oriented, relational-oriented, and values-based and moral leadership were associated with reduced workplace mistreatment. Additionally, ethical leadership was most strongly negatively associated with workplace mistreatment. In contrast, passive and destructive leadership were associated with increased workplace mistreatment [78]. This aligns with Bandura’s social learning theory, which posits that managers serve as key role models for their employees. According to this theory, the way managers handle situations, interact with others, and make decisions is reflected by employees in their own actions. However, for positive behaviors to be effectively imitated, employees must see that their organization rewards and reinforces these exemplary managerial practices [62]. From the practical perspective, case and intervention studies have found that transformational leadership training and supportive leadership training greatly increase employee well-being and job satisfaction across genders [79,80].

### 5.4. Practical Implications

The findings of this study highlight the importance of organizational awareness of the pervasive impact of lookism climate and its potential to foster workplace mistreatment. Certain measures aimed at reducing lookism climate and its associated behaviors could be beneficial in promoting a more inclusive and equitable workplace and in reducing occurrences of workplace mistreatment such as workplace incivility. While the difficulties of establishing and enforcing legislation and policies against appearance-based discrimination are extensive, certain organizational measures can help avoid the development of this subtle discrimination in the first place. For example, organizations could implement blind resume screening processes and standardized interview protocols to mitigate lookism bias in hiring and promotion decisions that may develop into a lookism climate. Blind/self-blind resume screening refers to the process of reviewing resumes without viewing demographic information such as gender, race, age, ethnicity, or candidate headshots, which may elicit biased decisions. Adapting these measures when evaluating job/promotion applications and resumes can ensure that decisions are being made as subjectively as possible [81]. Additionally, performance evaluation criteria and processes, along with other employment practices, should be clear and transparent. Research has demonstrated that when these processes are transparent, employees perceive a higher justice climate and a climate for inclusion, both of which are negatively associated with lookism climate [82].

Furthermore, steep vertical organizational structures may foster judgments based on appearance-related implicit biases, as higher management has limited contact with subordinates and thus little knowledge of their individual skills and competencies. Indeed, these tall hierarchical structures tend to favor agentic traits, prioritizing dominance, competitiveness, and assertiveness, as well as physical traits such as height and strength, which are traditionally associated with male leadership styles. This preference can create a bias not only towards hiring and promoting more physically fit and extroverted candidates but also towards promoting male candidates over equally qualified female candidates, thereby perpetuating gender disparities in leadership roles [83]. High power differentiation has also been found to increase interpersonal counterproductive conflict, such as knowledge hiding, a component of incivility [84]. By decreasing the height of organizational structures, organizations can mitigate decision-making based on implicit biases and reduce the likelihood of interpersonal conflict stemming from unhealthy competition. Organizations with steep hierarchical structures should maintain appropriate employee-to-manager ratios and implement regular manager–employee meetings and performance evaluations. This encourages consistent interaction between management and staff, helping to minimize biased judgments based on looks or other stigmatized characteristics.

### 5.5. Limitations

This study employed a self-report questionnaire to measure the explored variables, posing the risk of common method variance. However, implementing a completely anonymous questionnaire and utilizing behavioral experience scales rather than the self-labeling method likely mitigates some of these effects, such as those stemming from the social desirability bias [85]. To combat other sources of common method variance, we implemented additional procedural strategies as suggested by articles aimed at addressing endogeneity in vocational research. Such strategies included using reverse-scored items, carefully selecting control variables, and clearly separating dependent and independent variables in the questionnaire. However, to enhance the rigor of future studies, we can integrate instrumental variables into the questionnaire, randomize the order of the individual questions, or, preferably, employ an experimental design [85,86]. Additionally, our sample consisted of Italian workers across more than 13 different sectors. Thus, the generalizability of our results in other cultural contexts of non-specific work sectors is limited. Indeed, the mean values for lookism climate and workplace incivility were relatively low amongst our sample. Cultural differences in beauty standards and societal norms may vary results across more diverse samples and thus affect perceptions of lookism climate. Beyond the influence of beauty norms in cross-cultural perspectives of lookism climate, the distinction between collectivist and individualistic societies may play an even more critical role. In collectivist cultures, group conformity and harmony are often prioritized over individual appearance, potentially reducing the emergence of a lookism climate [1]. Furthermore, past research has found that lookism is particularly high in the hospitality and retail industries, while workplace mistreatment is particularly prevalent in the healthcare industry [87,88]. Additionally, appearance expectations fluctuate depending on the role, with lookism climate being more prevalent in front-facing positions, such as flight attendants, compared to roles with limited public interaction, like aircraft mechanics [88]. Therefore, future studies utilizing samples in specific sectors and roles are needed to see how the organizational processes influencing employee self-perceptions outlined in this study apply to different work environments. Moreover, the scale developed to measure lookism climate focuses exclusively on perceived organizational preferences for attractive employees and the organizational pressure to manage physical appearance. It does not address whether the organization views these attributes as linked to professional abilities. Consequently, the current measure primarily captures taste-based lookism. Future research should aim to create a two-dimensional scale that encompasses both taste-based and statistical lookism. This would help in determining whether employees believe the organization values attractiveness due to its perceived correlation with professional competencies or just based on biased preferences. Such an approach would enable a more nuanced exploration of lookism climate, particularly in sectors or roles where appearance may genuinely impact performance or productivity, such as those involving direct client interactions.

Lastly, the significant findings of this study were based on a cross-sectional design, which limits the ability to draw causal conclusions between lookism climate, exposure to workplace incivility, experiencing imposter syndrome, and having low perceived employability [89]. Alternatively, longitudinal studies, such as daily diary studies, can investigate the individual and collective effects of lookism climate on employees over time, therefore supporting causal relationships. These studies capture real-time experiences and fluctuations in variables, delineating temporal dynamics and sequences of events in the workplace, thereby offering high ecological validity [90]. Despite their advantages, longitudinal studies also present limitations. They are time- and resource-intensive, face challenges in maintaining participant attention and retention, and risk longitudinal bias due to participants’ heightened awareness of study variables and time lag miscalculations [90]. In summary, while a more rigorous study design would aid in drawing causal inferences between lookism climate, workplace incivility, impostor syndrome, and perceived employability, it would also come with its own set of limitations.

## 6. Conclusions

In conclusion, this study extends the literature on workplace discrimination by demonstrating the cascading effects of lookism climate on perceived employability, mediated by workplace incivility and imposter syndrome. For women, the process involves a serial mediation through workplace incivility and imposter syndrome, suggesting one explanation for the leak in women’s career progression through a ripple effect of organizational dynamics on self-perceptions. For men, a direct pathway from lookism climate to imposter syndrome provides a “short-cut” to lowered perceived employability. However, lookism climate still predicts experiencing workplace incivility for men. Consequently, this study underscores the importance of promoting positive workplace climates for the prevention of workplace incivility experiences. Additionally, the moderating effect of managerial competencies for women further underscores the importance of competent leadership and management of employees and their interactions with each other and the work environment in mitigating the negative impacts of lookism climate, experiencing workplace mistreatment, and in boosting occupational confidence. Future research should continue to explore these dynamics across different organizational contexts and across time to adequately demonstrate causality and develop more targeted and effective interventions. The nuanced understanding of these processes enhances our comprehension of workplace climates and mistreatment and their implications, contributing to the broader discourse on gender equality in the work environment.

## Figures and Tables

**Figure 1 behavsci-14-00883-f001:**
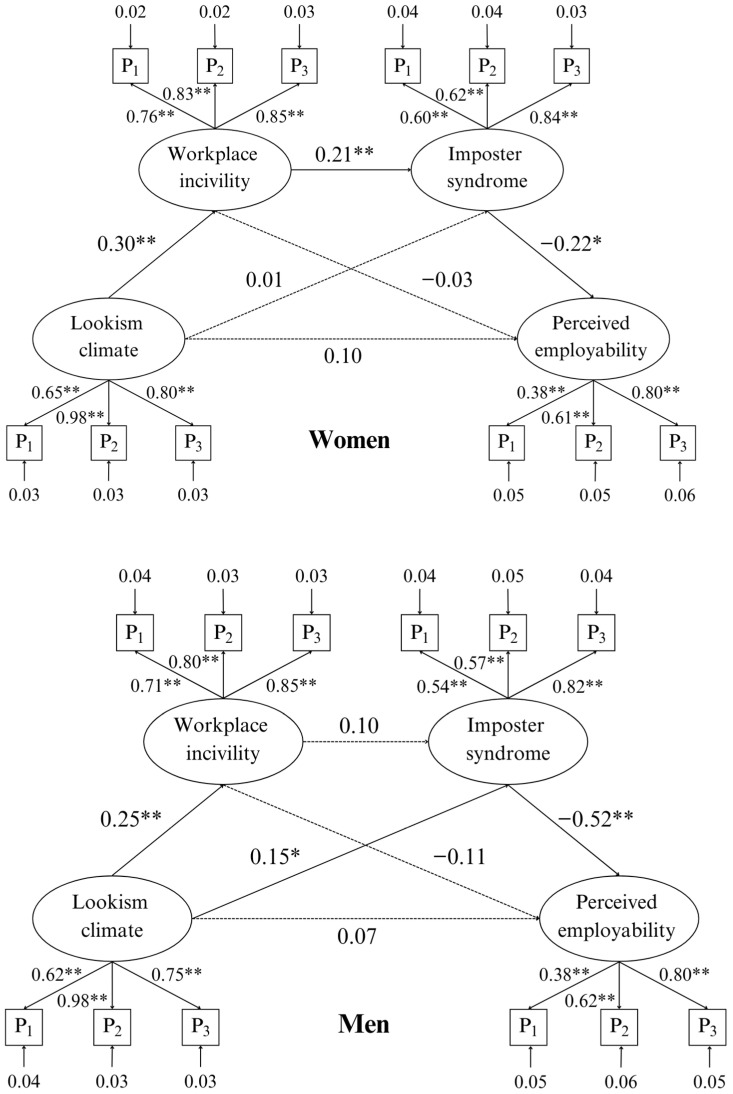
Multi-group serial mediation SEM path coefficients. Note: * indicates *p* < 0.05; ** indicates *p* < 0.01; — indicates significant path; --- indicates non-significant path.

**Figure 2 behavsci-14-00883-f002:**
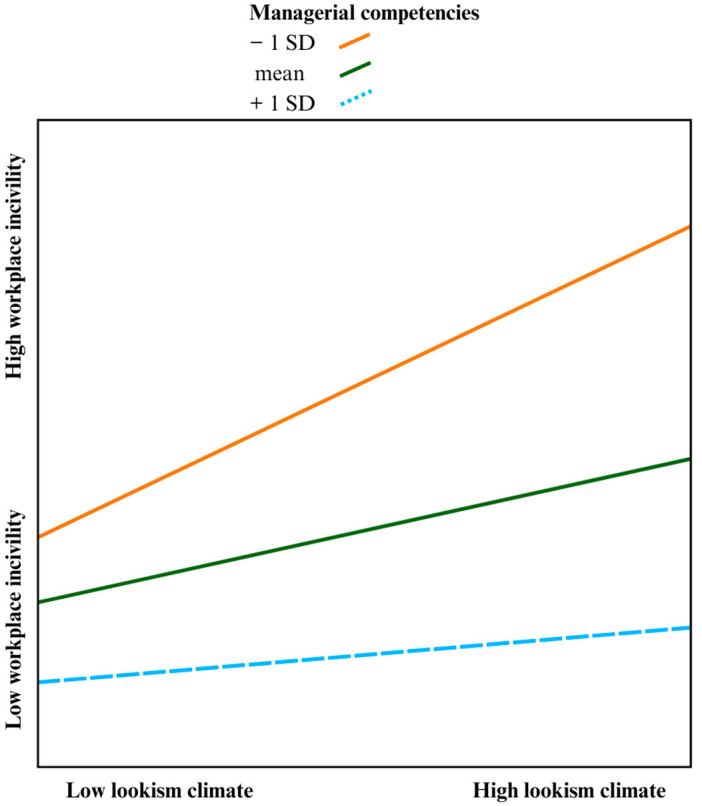
The moderating effects of managerial competencies on the indirect effect of lookism climate and workplace incivility exposure. Note: — indicates significant moderation; --- indicates non-significant moderation.

**Table 1 behavsci-14-00883-t001:** Means, standard deviations, and correlations with confidence intervals.

**Women**						
**Variable**	**M**	**SD**	**1.**	**2.**	**3.**	**4.**
1. Lookism climate	1.80	0.82				
			0.29 **			
2. Workplace incivility	1.80	0.69	[0.19, 0.39]			
			0.07	0.29 **		
3. Imposter syndrome	2.55	0.86	[−0.04, 0.18]	[0.18, 0.39]		
			0.03	−0.13 *	−0.12 *	
4. Perceived employability	3.41	0.57	[−0.07, 0.13]	[−0.23, −0.03]	[−0.21, −0.02]	
			−0.12 *	−0.35 **	0.03	0.19 **
5. Managerial competencies	3.02	0.89	[−0.22, −0.02]	[−0.45, −0.25]	[−0.9, 0.13]	[−0.22, −0.02]
**Men**						
**Variable**	**M**	**SD**	**1.**	**2.**	**3.**	**4.**
1. Lookism climate	1.85	0.82				
			0.23 **			
2. Workplace incivility	1.64	0.60	[0.11, 0.35]			
			0.16 **	0.15 *		
3. Imposter syndrome	2.48	0.80	[0.06, 0.28]	[0.03, 0.26]		
			−0.02	−0.21 **	−0.29 **	
4. Perceived employability	3.59	0.56	[−0.15, 0.11]	[−0.31, −0.09]	[−0.39, −0.18]	
			−0.18 **	−0.46 **	−0.03	0.17 **
5. Managerial competencies	3.22	0.81	[−0.29, −0.07]	[−0.56, −0.36]	[−0.15, 0.09]	[0.05, 0.30]

Note: values in square brackets indicate the 95% confidence interval for each correlation; * indicates *p* < 0.05; ** indicates *p* < 0.01.

## Data Availability

The datasets generated and analyzed during the current study are available from the corresponding author upon reasonable request.

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
