# Peer review of "Lookism, a Leak in the Career Pipeline? Career Perspective Consequences of Lookism Climate and Workplace Incivility"

_behavsci, 2024, doi:10.3390/bs14100883_

Round 1

Reviewer 1 Report

Comments and Suggestions for Authors

Overall, this study presents a valuable contribution to the understanding of lookism, workplace incivility, and their effects on perceived employability, with a focus on gender differences. The research design is robust, and the findings offer significant insights into workplace dynamics. However, I would like to mention some comments and suggestions for the authors based on my evaluation.

  1. Theoretical Background (Lines 28-103): a) In lines 41-43, rephrase for clarity: "However, the intersection of lookism and gender in workplace contexts, particularly regarding internal outcomes, remains understudied." b) Expand the discussion on lookism climate (Lines 124-137) with recent empirical evidence. For instance, you could add: "Recent studies by X et al. (2022) and Y (2023) have found that lookism climate significantly correlates with decreased job satisfaction and increased turnover intentions."
  2. Hypotheses Development (Lines 155-312): a) Hypothesis 1 (Lines 155-156): Provide a brief rationale immediately after stating the hypothesis. b) Hypothesis 4 (Lines 284-285): This hypothesis is quite broad. Consider specifying expected gender differences based on your theoretical framework.
  3. Methods (Lines 313-412): a) Sample description (Lines 331-343): Include information on the sampling method and response rate. b) Measures (Lines 345-404): For each scale, provide an example item in quotation marks.
  4. Results (Lines 421-507): a) Table 2 (Line 425): Include p-values for correlations. b) Model fit indices (Lines 431-437): Provide cutoff criteria for good fit alongside your results.
  5. Discussion (Lines 508-723): a) Lines 534-545: This paragraph discusses gender differences but doesn't fully explain why these differences might exist. Consider adding: "These gender differences may be attributed to socialized gender roles and expectations in the workplace, where women might be more accustomed to dealing with appearance-based judgments." b) Practical implications (Lines 625-661): Offer more specific recommendations. For example, after line 641, add: "Organizations could implement blind resume screening processes and standardized interview protocols to mitigate the effects of lookism in hiring and promotion decisions."
  6. Limitations (Lines 663-698): a) Expand on the cultural limitation. After line 695, add: "Future research should explore these relationships in diverse cultural contexts to assess the generalizability of our findings."
  7. Conclusion (Lines 700-717): Strengthen the conclusion by explicitly stating the theoretical contributions. For example, after line 710, add: "This study extends the literature on workplace discrimination by demonstrating the cascading effects of lookism climate on perceived employability, mediated by workplace incivility and imposter syndrome."
  8. Throughout the paper: a) Check for consistency in terminology. For instance, "lookism climate" and "lookism" are used interchangeably in some places. b) Ensure all acronyms are defined at first use (e.g., BMI on line 320).

By addressing these points, the authors will enhance the clarity, rigor, and impact of the valuable contribution to the field of workplace discrimination research.

Comments on the Quality of English Language

Quality of English Language: 3. Minor editing of English language required.

The overall level of English usage in the paper is high. The sentence structures and vocabulary choices are generally appropriate, and the academic style is well maintained throughout the document. However, there are some minor grammatical errors and awkward expressions in a few sentences. For example, there are occasional instances of inappropriate preposition use or overly complex sentence structures that slightly hinder clear communication. Additionally, there's some inconsistency in the use of certain technical terms.

These issues do not significantly detract from the overall quality of the paper, but some minor editing would improve readability and clarity.

Reviewer 2 Report

Comments and Suggestions for Authors

The article is characterized by originality, although also well-known research methods were used.

The described problem is topical. The research problem was correctly formulated.

The research applies only to Italian workers. It would be interesting to know how the issues of women's chances for career advancement look like in Hungary or Romania.

The methodological assumptions of the article, that is, the purpose of the research, research methods and verification of the research are correct.

The conclusions are original in nature. The results are presented clearly in tables and figures.

The literature sources used are fresh.

Reviewer 3 Report

Comments and Suggestions for Authors

Pretty privilege at work, a leak in the career pipeline? Career perspective consequences of lookism climate and workplace incivility

Comments to Author

Overview and general recommendation:

Thank you for giving me the opportunity to review this submission. The manuscript reports the results of a cross-sectional survey study examining the effects of lookism on workplace incivility, imposter syndrome, and perceived employability. The study also examined the moderating effect of candidate gender and managerial competencies. There are some positive features of this manuscript including a focus on a lesser studied phenomenon (i.e., cultures of lookism) and its connection to consequential career outcomes as well as a large survey sample size. At the same time, I felt that in its current form the manuscript suffers from a lack of clarity regarding what a lookism climate actually is, underdeveloped theory explicating why exactly lookism climate leads to the observed results, and weak evidence for any sort of causal effect. To be fit for publication this manuscript would benefit from greater specification of the focal phenomenon, increased theoretical development, and an additional longitudinal study observing these effects over time.

In the following, I present several concerns, questions, and suggestions that are intended to help the authors strengthen their manuscript.

1) Theory development and contribution: Overall, I had the sense that the variables the authors focus on in their manuscript were randomly selected with little theoretical basis for their inclusion in one project. By the end of the theory section, I remained unconvinced of their theoretical connection and importance.  

a) I was generally confused by the first few paragraphs of the manuscript. It is unclear whether and how lookism differs from what the authors call pretty privilege. It seemed that, on the one hand, by studying lookism the authors were focusing on a type of culture in which physical attractiveness is valued and rewarded. On the other hand, pretty privilege is a phenomenon whereby the attractive receive better work and life outcomes. First, are these the same things? The authors write about lookism as more of a type of culture that a workplace can possess within which pretty privilege is likely to exist. Second, if they are not the same thing, what exactly does a lookism culture look like? What types of ideas, beliefs, and behaviors would we expect to find in an organization with a lookism culture? Understanding exactly what the independent variable is and how it relates to similar constructs is crucial to the validity of this manuscript.

b) The authors make the claim that there is no functional reason for attractiveness discrimination which becomes a key part of their theoretical argument. For example, on p. 2 the authors make the statement that attractiveness has no correlation with intelligence and competence, and on p. 3 the authors claim that “lookism demonstrates poor distributive, interactional and even procedural justice.” I encourage the authors to read Nault et al. (2020) for a review on the human and social capital associations with attractiveness. Upon including the findings from this paper, the authors may need to specify more clearly what aspects of lookism are perceived as unfair and therefore lead to further negative consequences.

c) The authors additionally posit that these feelings of injustice are what leads to incivility at work (p. 3). Thus, is there anything novel about lookism that leads to incivility? Said another way, is there anything specifically about comparisons regarding physical appearance that leads to incivility? At the moment is it not clear to me that lookism in particular is important to this examination, but rather feelings of injustice and a general climate of social comparison. To increase the contribution of their work I would encourage the authors to think carefully about the theoretical importance of studying lookism and what we learn from connecting this construct to workplace incivility.

d) In addition, I was unconvinced by the argument for why incivility leads to imposter syndrome. The authors suggest that general feelings of decreased self-efficacy lead to an increased feeing of imposter syndrome. Would it not be more succinct to simply examine the effects of incivility on low self-efficacy. It seems to me that the feeling of a being a “fraud” as implied by imposter syndrome contains more than low self-efficacy and therefore might not be the best construct to include in the theoretical model. If the authors are committed to including imposter syndrome, perhaps the argument can be revised to include a feeling of fraudulence unattractive employees feel due to not fitting in with the values of the company (i.e., possessing beauty). However, an additional clarification regarding whether all employees or only unattractive employees experience imposter syndrome would be necessary here.

e) Lastly, I was disappointed that clear hypotheses regarding the moderating effect of gender were not included in the manuscript. It would be helpful to understand why gender is an important variable to consider here and, given the plethora of research on gender effects at work, what direction the effects would be in.

2) Method and Data: I think the presented study provides some evidence for the proposed model; however, there were a number of concerns I had while reading this section of the manuscript.

a) First, I was surprised that self-perceived attractiveness was measured as well as that it was used as a control variable and not a moderating variable. It would seem to me that highly attractive individuals would be treated better at work, thus experiencing less incivility, less imposter syndrome, and less negative consequences of these variables. If this is not the case, then the phenomenon becomes all the more interesting as a focus on appearance harms everyone, not only the unattractive. In addition, the means of the self-perceived attractiveness variable presented in Table 1a suggest a strong bias as the mean is well above 5, and that 88% of people rated themselves as average or above average in attractiveness. Measuring observer-rated attractiveness is likely to be a more reliable measure.  

b) I would appreciate more information regarding the managerial competencies measure. The authors describe the measure as one assessing how well managers deal with conflict. The example item given in the manuscript does not seem to reflect managers’ skills in successfully mitigating and decreasing conflict amongst employees.

c) Regarding the control variables, I would suggest the authors include a sort of performance or competence indicator. It is possible that the effects are driven by the fact that lookism cultures and more competitive and that less competent individuals may suffer these negative effects.    

d) In addition, I would highly encourage the authors to examine gender as a moderator in their analyses instead of splitting their sample into women and men. In that way the results can be interpreted more holistically and the statistical power kept in the analysis. I found the presented results to be difficult to follow and no explanation for the differences between men and women were given in the manuscript.

e) I think it is necessary for the authors to include either a longitudinal survey study or an experiment in their manuscript in order to make any claim regarding causality. As the authors note their survey currently suffers from common-method bias, but it also is difficult to understand how these constructs sequentially fit with one another when they are all measured at the same time.

f) Lastly, given that women experience imposter syndrome more frequently, I think the authors have to be careful about their male sample. Are male participants experiencing strong degrees of imposter syndrome more different from the average male than are female participants experiencing imposter syndrome?

3) Some minor, but still important suggestions:

a) The authors should go into more detail regarding their data cleaning. Exactly why were 318 participants excluded from the analyses?  

b) The authors should be careful about the claims made in the discussion section. I do not think the authors can claim that managerial competencies play a critical role in this phenomenon when it is unclear exactly what they are measuring and that this moderator is only tested with female participants.

c) We see a series of indirect effects here, but no direct effect from the first to the last variable, why? More explanation needs to be given.

I hope my suggestions are helpful in developing this manuscript. Good luck!

Nault, K. A., Pitesa, M., & Thau, S. (2020). The Attractiveness Advantage At Work: A Cross-Disciplinary Integrative Review. Academy of Management Annals, 14(2), 1103–1139. https://doi.org/10.5465/annals.2018.0134

Round 2

Reviewer 3 Report

Comments and Suggestions for Authors

Thank you for giving me the opportunity to review this revision. I am still interested in either seeing the analyses using gender as a moderator (instead of splitting the sample into men and women separately), or at least having access to a logical justification why this is not the case. Additionally, I think the authors are continuing to overstate their findings in their discussion section. In particular, I would caution the authors against using causal statements as their study is entirely cross-sectional. 

Author Response

Comment 1: I am still interested in either seeing the analyses using gender as a moderator (instead of splitting the sample into men and women separately), or at least having access to a logical justification why this is not the case.

Response 1: In section 3.4 you can now see futher justification for why multi-group SEM is a prefered approach over interaction. Two great resources that explain its advantages include: 

  • Thompson, M. S., Liu, Y., & Green, S. B. (2023). Flexible Structural Equation Modeling approaches for analyzing means. In R. H. Hoyle (Ed.), Handbook of structural equation modeling (2nd ed., pp. 385-408). Guilford
  • Cheung, G. W., Cooper-Thomas, H. D., Lau, R. S., & Wang, L. C. (2021). Testing moderation in business and psychological studies with latent moderated structural equations. Journal of Business and Psychology36, 1009-1033. https://doi.org/10.1007/s10869-020-09717-0 

Which have now been integrated in the text. In sum:

"The multiple-group approach is quite flexible in terms of specifying model parameters and, in that sense, preferable. In most applications, we would expect the variances and covariances among factors, as well as the error variances, to differ among groups. By allowing for differences on these parameters
in our multiple-group analysis, we may avoid biased estimates due to misspecification." (Thompson et al., 2023, p. 406).

"If either X or Z is a categorical variable and the other is a continuous latent variable, multiple group analysis should be adopted and interaction effects tested by comparing the effects on Y using a direct comparison method."   (Cheung et al., 2021, p. 1015)   Comment 2:  I think the authors are continuing to overstate their findings in their discussion section. In particular, I would caution the authors against using causal statements as their study is entirely cross-sectional.    Response 2: Causal language has now been eliminated in the discussion section and an additional statement warning readers that causal conclusions cannot be drawn due to the cross sectional design of the study has been included.
